# Putting self at stake by telling a story: Storyteller's narcissistic traits modulate physiological emotional reactions to recipient's disengagement

Emmi Koskinen[1]*, Pentti Henttonen[1,2], Ville Harjunen[1], Elizabeth Krusemark[3], Matias Piispanen[4,5], Liisa Voutilainen[6], Mariel Wuolio[4], Anssi Peräkylä[4]

**1** Faculty of Medicine, University of Helsinki, Helsinki, Finland, **2** Helsinki Institute for Social Sciences and Humanities, University of Helsinki, Helsinki, Finland, **3** Millsaps College, Jackson, Mississippi, United States of America, **4** Faculty of Social Sciences, University of Helsinki, Helsinki, Finland, **5** Department of Neuroscience and Biomedical Engineering, Aalto University, Espoo, Finland, **6** School of Educational Sciences and Psychology, University of Eastern Finland, Joensuu, Finland

* emmi.ek.koskinen@helsinki.fi

**Data Availability Statement:** Summary statistics from physiological data and self-reports are made available in Open Science Framework: https://osf.

## Abstract

Telling a story to a disengaged recipient induces stress and threatens positive self-image. In this study, we investigated whether storytellers with overly positive and fragile self-images (e.g., individuals with grandiose and vulnerable narcissism) would show heightened behavioral, emotional, and psychophysiological reactivity to recipient disengagement. Building on Bavelas, Coates, and Johnson [1] we conducted a conversational experiment instructing the participants to tell about a "close call" experience to a previously unknown co-participant. We modified the co-participant's level of interactional engagement by asking them either to listen to the story carefully or to simultaneously carry out a counting task that distracted them from the content of the story. We found that the distraction condition was unrelated to the storytellers' narration performance, but a significant positive association was found between the story-recipients' observed lack of affiliation and the tellers' narration performance. The distraction of recipients was also associated with increased self-reported arousal in the tellers, indicating disengagement-induced stress in the tellers. Moreover, tellers higher in grandiose narcissism reacted with higher skin conductance response to disengagement, and vulnerable narcissism was associated with higher heart rate during narration in general. Our experiment thus showed that grandiose narcissists are emotionally sensitive to their co-participants' disengagement.

## Introduction

### Storytelling, face, and narcissism

Storytelling is a primordial site for humans to share emotions, experiences, and create "empathic moments" in interaction [2, 3]. In everyday interactions, we are frequently telling

io/v7f63/?view_only=
e08509ad662f405b92183f74d084ad7a Audio and
video data are not shared in order to protect the
anonymity of the research participants.

**Funding:** This work was supported by the
Research Council of Finland, project numbers
319113 and 320248 (https://www.aka.fi/en/), and
the Society of Swedish Literature in Finland
(https://www.sls.fi/en). Open access funded by
Helsinki University Library. The funders had no role
in study design, data collection and analysis,
decision to publish, or preparation of the
manuscript.

**Competing interests:** The authors have declared
that no competing interests exist.

each other stories about events that have happened to us, be they big or small. In general, recipients of these kinds of stories are expected to display *affiliation* by endorsing the teller's affective treatment of the events [4]. If recipients do not show affiliation in a relevant place of the telling, their actions can be understood as unempathetic [2, 4, 5]. The tellers also monitor recipients' responses very closely and adjust their own actions accordingly if the recipient seems uninterested or disengaged. So far, it has remained unstudied whether people differ in their sensitivity to others' interactional disengagement and whether traits associated with seeking admiration and defensive strategies such as narcissism [6] make people more prone to the socio-emotional perturbations caused by others' interactional disengagement. In this study, we therefore investigated whether grandiose and vulnerable narcissism influences storytellers' behavioral, emotional, and psychophysiological reactions to disengaged story-recipients. We expect narcissistic individuals to be especially sensitive to (lack of) recipient feedback and thus provide a special 'window' to these interactional and emotional processes.

Observing that the recipient is not engaged in the story can be unsettling and compromise the teller's linguistic performance. In a seminal study, Bavelas, Coates, and Johnson [1] asked participant A in a dyad to tell about a "close call" in their life to an unknown co-participant B. In half of the dyads, participant B was told to listen to the story, and in half of the dyads participant B was told to count the words that begin with the letter T. In the manipulated condition where the listener was disengaged because of the counting task, the overall performance of the storyteller dropped significantly. The tellers did particularly poorly at what should have been the "dramatic conclusion" ([1], p. 950) of the story. The tellers' linguistic performance can therefore be influenced by the recipients' engagement. More recent research shows, however, that also the autonomic nervous system activity of a storyteller can be affected by recipient's lacking feedback and displays of affiliation. Peräkylä and colleagues [5], for example, found that lack of recipients' affiliation increased the storytellers' level of sympathetic arousal (as measured by skin conductance responses), whereas expressions of affiliation decreased the tellers' sympathetic drive. The authors interpreted the results in relation to the concept of *face* [7]: in telling a story, the speaker is putting something of him/herself "out there" for others to judge, and so the expected, affiliative response from the recipient leads to relaxation (maintaining face), while the lack of it increases anxiety (losing face) (see [5], p. 306). It is likely, however, that individuals vary in their propensity for the maintenance of face in interaction based on their personality traits and other characteristics [8–10]. Especially individuals with narcissistic traits should exhibit sensitivity to the threat of losing one's face during interpersonal interactions due to their pronounced emotional reactivity to self-threatening social feedback [11] and need for admiration [6]. Until now, however, no direct evidence has been reported to demonstrate the role of narcissistic traits in amplifying the emotional reactions to others' disengagement.

As individuals with narcissistic personality traits have a heightened preoccupation with the self [12] narcissism can cast light into the processes of losing and maintaining face in interaction. In the context of storytelling, this could mean that storytellers with narcissistic traits are emotionally more sensitive to (lack of) recipient affiliation. Considering the variability in narcissistic traits, however, the actual behavioral reactions of narcissistic storytellers are likely to be more complex. Researchers have typically differentiated narcissism into two types: *grandiose* (overt) and *vulnerable* (covert) narcissism [13]. In grandiose narcissism, individuals have higher self-esteem, extraversion, lower symptoms of anxiety and depression, and lower shame proneness [14, 15], whereas individuals with vulnerable narcissism exhibit lower self-esteem, higher proneness to shame and anxiety, and greater difficulties with emotion regulation [16, 17]. Grandiose individuals can be more sensitive to achievement failure, whereas vulnerable individuals can be more sensitive to interpersonal rejection [18], as in storytelling contexts.

Individuals with grandiose traits react with aggression in response to social exclusion [19–21], but do not necessarily report emotional distress concurrent with increased physiological arousal and facial muscle activity associated with negative valence [11, 22, 23]. There are theories that grandiose narcissistic individuals use their social interactions mainly as opportunities to self-enhance, effectively seeking admiration instead of interpersonal affiliation [24, 25]. Furthermore, grandiose narcissism is negatively related to emotion contagion (e.g., the transmission of emotions from one individual to another: [26], which may buffer individuals from the negative emotions of others. It is therefore possible that grandiose narcissists could be indifferent to a lack of affiliation, which prevents instability of their own emotional responses (see also [27]). In contrast to grandiose narcissism, vulnerable narcissism is associated with greater distrust of others, social inhibition, and negative emotionality [28]. For this reason, vulnerable narcissists may interpret others' behavior as more malevolent and threatening than do grandiose narcissists [29]. It is therefore possible that individuals with higher levels of vulnerable narcissism may experience lack of recipient feedback as more threatening than grandiose narcissists. In the current study, we aimed to achieve a better understanding of the relation of narcissistic personality traits and the reactivity to other's interactional engagement.

Considering that individuals with narcissistic personality traits tend to provide biased reports regarding their own abilities, emotional reactions, and individual experiences [30–32], physiological assessments have been suggested to provide more objective indices of intrapersonal processes and emotional responses [22].

## Narcissism and psychophysiology

As narcissistic individuals show reactivity to interpersonal rejection, it is expected that narcissistic individuals respond physiologically under circumstances involving face threats during social interactions. Experimental research utilizing manipulations involving interpersonal rejection, social-evaluative stressor tasks, and achievement failure reveals a general pattern of physiological reactivity, but the findings are equivocal due to the variety of tasks and different methods of narcissism assessment. Individuals with grandiose traits show sensitivity to social exclusion and interpersonal stressors, marked by higher levels of psychophysiological and neuroendocrine reactivity [33–35] and stronger facial muscle activity associated with negative emotional states [11, 36]. Conversely, there is evidence that individuals with grandiose traits exhibit reduced electrodermal activity in response to social threats and aversive cues [37, 38], which supports the previously mentioned hypothesis that grandiose individuals may be less reactive to interpersonal face threats. Due to the somewhat contradictory findings, there is a need for further studies. Studies of psychophysiological reactivity in vulnerable narcissism are limited but indicate that vulnerable traits are associated with increased cardiovascular reactivity and slower recovery following rejection [37] and greater stress-induced respiratory-sinus arrhythmia (RSA) reactivity [17]. Taken together, these findings demonstrate a tendency to experience physiological reactivity to social stressors for grandiose and vulnerable individuals. In addition to this evidence, it is also useful to consider the broader interpersonal contexts that elicit physiological reactivity for narcissistic individuals in order to develop specific hypotheses regarding the relationship between narcissism and physiological arousal in response to recipient's disengagement in the context of storytelling.

Overall, the behavioral and psychophysiological evidence illustrates a general pattern of emotional and physiological reactivity to social stressors and face threatening events for individuals with narcissistic traits. It is important to note, however, that these studies are limited to laboratory tasks or self-reports of daily events outside of the lab. Utilizing more natural experimental settings provides two benefits to understanding face concerns in social interaction and narcissistic personality: examining conversational interactions provides a) an ecologically valid

social situation in which face concerns are authentic, and b) an experimental design that manipulates the degree of engagement from a co-interactant.

## Current study

Our experimental storytelling setting was based on the study mentioned above by Bavelas, Coates, and Johnson [1]. We used the same letter counting manipulation–with the exception of changing the letter T to the letter K to be more suited in the Finnish language context–to lower the recipients' engagement. We used external observers to rate the tellers' performances and measured their trait narcissism scores (grandiosity and vulnerability), psychophysiological arousal (skin conductance, SC and heart rate, HR), and self-reported valence and arousal ratings during the storytelling task. The study was pre-registered in Open ScienceFramework [39]. All statistical models and tests corresponded to those posited in the preregistration document.

In order to investigate interindividual variation in narcissism and to capture both the grandiose and vulnerable traits of narcissism, we asked the participants to complete two short self-report inventories: brief version of the Narcissistic Personality Inventory (NPI-13; [40]) and the vulnerable narcissism subscale from the Super Brief Pathological Narcissism Inventory (PNI: [41, 42]). We used brief versions of the inventories due to time constraints (see section on trait measures below).

## Hypotheses

Our first hypothesis (H1) builds on Bavelas et al. [1], who showed that distracting a story recipient from the content of the storytelling was associated with problems in the quality of the teller's narration. H1 thus states that the *distracted recipient condition* (counting task), as compared to the *non-distracted recipient* condition, is associated with worse performance of the teller as evaluated by independent raters.

Our second hypothesis (H2) builds on the findings by Peräkylä et al. [5], who showed that lack of recipient affiliation during storytelling was associated with higher psychophysiological arousal in storytellers. H2 thus states that the *distracted recipient* condition (counting task) is associated with more negative valence ratings and increased self-reported and autonomic arousal in the teller than in the *non-distracted recipient* condition.

Our third hypothesis (H3) builds on the evidence that both grandiose and vulnerable narcissism are associated with emotional reactivity to social stressors and interpersonal rejection and behavioral patterns that suggest sensitivity to others' affiliative behaviors. H3 thus states that the teller's level of trait narcissism moderates the effect in the teller's performance, their self-reported affect, and their psychophysiological reactivity in the *distracted recipient* condition. Specifically, we expect that both grandiose and vulnerable narcissistic traits predict greater drop in the teller's performance, and higher autonomic and self-reported affective response to recipient's disengagement. In addition, given that vulnerable narcissism is associated with higher negative affect and physiological reactivity to interpersonal rejection than grandiose narcissism, we expect that individuals with high levels of vulnerable narcissism will exhibit stronger autonomic reactivity and self-reported affect in response to distracted recipient than those with high levels of grandiose narcissism.

## Materials and methods

### Participants

The participant sample consisted of 84 volunteers randomly assigned to 42 same-sex dyads (29 female-female, 13 male-male; see Table 1 for full information of the participant

**Table 1. Demographics of the participants and experimental variables.**

| Group | Condition | $n$ | Age (y.) | | NPI-13 Total (13–65) | | SB-PNI-V (0–30) | |
|---|---|---|---|---|---|---|---|---|
| | | | $M$ | $SD$ | $M$ | $SD$ | $M$ | $SD$ |
| Tellers | Control | 20 | 29,15 | 11,29 | 22,90 | 5,61 | 11,05 | 4,32 |
| | Distraction | 22 | 27,14 | 7,50 | 26,18 | 8,74 | 13,09 | 5,66 |
| Recipients | Control | 20 | 30,35 | 10,62 | 28,35 | 8,52 | 13,10 | 6,22 |
| | Distraction | 22 | 26,68 | 8,68 | 26,50 | 8,02 | 12,23 | 4,46 |
| All | All | 84 | 28,26 | 9,51 | 26,00 | 7,95 | 12,38 | 5,19 |
| Group | Condition | $n$ | Story potential * (1–5) | | Storytelling quality* (1–5) | | Recipient affiliation (1–5) | |
| | | | $M$ | $SD$ | $M$ | $SD$ | $M$ | $SD$ |
| Tellers* or recipients | Control | 20 | 3,08 | 0,90 | 3,48 | 0,81 | 3,35 | 1,10 |
| | Distraction | 22 | 2,70 | 0,88 | 3,35 | 0,72 | 3,33 | 0,93 |
| | All | 42 | 2,88 | 0,90 | 3,41 | 0,75 | 3,34 | 0,99 |
| Group | Condition | $n$ | SAM valence (1–9) | | SAM arousal (1–9) | | SAM dominance (1–9) | |
| | | | $M$ | $SD$ | $M$ | $SD$ | $M$ | $SD$ |
| Tellers, post-experiment | Control | 20 | 7,58 | 1,21 | 2,73 | 1,52 | 4,95 | 1,00 |
| | Distraction | 22 | 7,23 | 1,31 | 3,82 | 1,84 | 4,68 | 1,64 |
| | All | 42 | 7,27 | 1,33 | 3,57 | 1,80 | 4,99 | 1,49 |

demographics). The sample was recruited from university mailing lists as well as via live recruitments at the university premises. Data collection was conducted between 30th September and 7th of December in 2021. Mean age of participants was 28.26 years (SD = 9.51, range: [18–67]). The roles (teller or recipient) were assigned based on the flip of a coin. Participants were told that the experiment concerns the participants' personality traits relating to self-image and physiological responses during storytelling. Participants received gifts valued 5€ as compensation. Condition was pseudorandomized by the stimulus computer at the onset of the experiment, with 22 dyads assigned into distraction condition and 20 remaining as controls. Participants provide informed consent and signed a written consent form. The conversational experiment and its procedure were accepted by the University of Helsinki Ethical Review Board in Humanities and Social and Behavioral Sciences (statement 8/2021).

The required sample size was calculated using analytical and simulation based power calculation approaches of the "pwr", "paramtest" and "InteractionPower" packages in R. In the pre-registration, we reported power calculations based on a large effect of d = .80 reported by Bavelas and colleagues [1] when examining the difference in narration performance between the distracted vs. non-distracted recipient condition. The analytical power analyses revealed that to detect such a large effect with 80% statistical power, 26 individuals (i.e., storytellers) per condition were required. Then, simulation analysis was performed to determine the required sample to reliably detect smaller main effects with 80% statistical power. Based on the results, a sequential analysis strategy [43] was used to decide whether to terminate the data collection before reaching the maximum sample size of 120 individuals. The data collection was terminated when 40 dyads (20 tellers) in both conditions were measured, partially due the difficulties of recruiting new dyads during the COVID-19 pandemic. Because a smaller-than-intended sample was collected, a sensitivity power analysis was carried out to determine the minimum effect size that could be reliably detected with 80% statistical power, based on the N = 20 (per condition) and a .05 alpha level. This sensitivity analysis was carried out for the interaction effect between distraction condition and the NPI-13 score, that was the primary

effect of interest in the study. The analysis revealed that with a sample of 20 per condition, alpha level of .05, and target power of 80%, the minimum detectable effect size was r = .58 (See S1 Supplementary material). Summary statistics from physiological data and self-reports are made available in Open Science Framework [39].

## Procedure

After giving informed consent and affixing the measuring equipment, participants were led into the laboratory, where they sat two meters apart in chairs facing each other (See Fig 1). Both participants received individual instructions from monitors not visible to each other. The participants were video recorded with three cameras, two facing each participant and the third giving a side view of the interactants. Two lavalier microphones were used for voice recording. The paradigm was programmed and run with Octave PsychToolBox 3 [44]. The instructions were presented with dual 24" 1920x1080 monitors in a diagonal layout. Participants sat diametrically with a table between them. Low screens obstructed both participants' hands from each other, where the teller had a functional computer mouse, whereas the recipient had an inactive mouse and a response button.

The experiment started with a baseline rest period of 3 minutes, where the participants were instructed to relax. Afterwards, the dyad members assigned to teller conditions were requested to tell a personal story about a close-call situation, where something bad almost happened to them (e.g. a skiing or riding accident, losing an important file from their computer) but in the end everything turned out ok. They were asked to describe the event in as much detail as possible. Remaining dyad members were, according to condition, requested to either "listen carefully to the story" or to "count all the letters starting with the letter 'K' in the teller's speech", by pressing a button with their dominant hand. The rate of correct detections was remotely monitored by a researcher during the experiment. All participants qualified as having performed the task as instructed.

The teller indicated readiness to start and finish the story by pressing a button. In case of prolonged telling, the screen indicated that the story is long enough at the 4 minute mark and advised to stop the task at the 5 minute mark. Finally, a 3 minute post-test baseline rest was recorded. Afterwards, participants were told that this is the end of the experiment and were offered the opportunity to repeat the task with reversed roles. The experiment commenced by

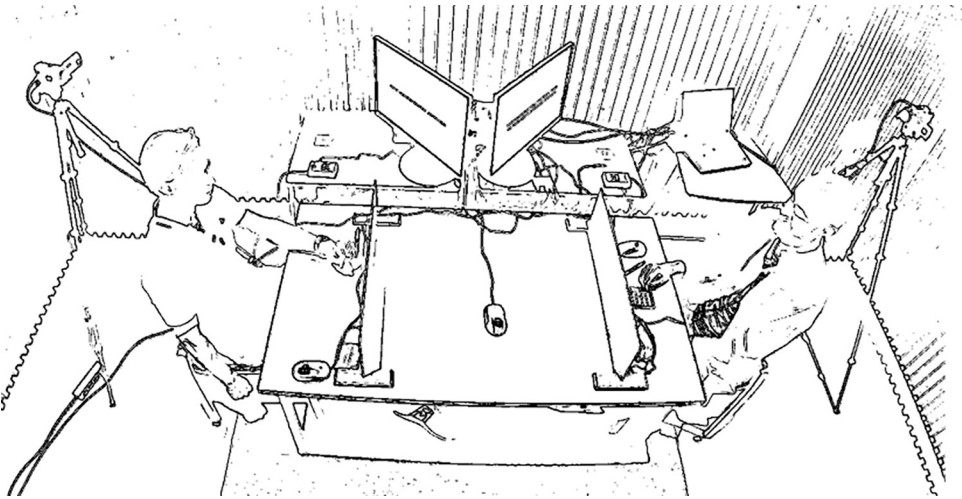

**Fig 1. Anonymized picture of the experimental setup.**

participants filling out the questionnaires. After removing the measurement devices, participants were individually debriefed.

## Physiological data acquisition and pre-processing

**Skin conductance and electrocardiogram data.** Physiological data from both participants were recorded with individual Nexus-10 devices [45]. ECG was sampled at 512 Hz with two wet Ag/AgCl electrodes placed according to the modified Lead II configuration. Skin conductance was measured at 32 Hz with a constant voltage of 0.5 V between two dry Ag/AgCl electrodes affixed the palmar middle phalanges of the index and middle fingers of the non-dominant hand. The recordings were synchronized with each other and the video recording using analog triggers from the stimulus computer which were routed to all devices.

**Preprocessing of physiological data.** SC data was preprocessed using Ledalab toolbox for Matlab [46]. Original time series data were downsampled to 16 Hz and smoothed with an adaptive filter. Data were visually inspected, and short (<10 seconds) movement artifacts were manually interpolated. Five participants' SC data during the task were discarded, due to nonresponse/flat signal (2) and excessive artifacts/bad quality signal (2) and an experimental error (1). Phasic component of the signal was extracted from the time series using continuous decomposition analysis [47]. ECG data was analyzed by low pass, high pass and band pass (mains noise) filtering and detection of R-peaks using the standard QRS detection algorithm [48]. Resultant inter-beat-interval series was manually screened for artifactual and ectopic beats. From the screened time series, average heart rate (HR, beats per minute) was calculated for the periods of interest. Average levels of derivative signals during the telling task and both baselines were computed. As an index of individually normalized activation, the mean phasic activation during the telling was divided with the activation during baseline. As two participants had excessive movement artifacts during the pre-experiment baseline, we used the post-experiment baseline as the rest state metric. The resulting parameter thus describes individual activation in relation to the rest state afterwards.

**State measures.** Prior to and after the experiment, participants reported their emotional state valence (negative vs. positive), arousal (high vs. low) and dominance (dominant vs. submissive) on a nine-point scale according to the Self-Assessment Manikin (SAM) scales [49]. As two dyads' initial scores were not obtained due to experimental error, only the post-experiment scores were used in the analysis.

**Trait measures.** After the experiment, participants were given a short questionnaire. We utilized two self-report measures for the grandiose and vulnerable dimensions of narcissism, the Narcissistic Personality Inventory (NPI-13; [40, 50]) and the Super Brief Pathological Narcissism Inventory (SB-PNI; [41, 42]. Item translations were adapted from a prior validation study, where both scales were demonstrated to be internally and externally valid instruments in a Finnish population comparable to the sample used in this study [51].

**Grandiose narcissism.** Raskin and Hall [50] developed the Narcissistic Personality Inventory (NPI) for measuring narcissistic traits. The NPI remains the most commonly utilized measure of narcissistic traits as well as a prominent measure of grandiose narcissism [52]. We utilized the brief, 13-item NPI that includes 3 subscales pertaining to the three-factor solution: Leadership/Authority, Grandiose Exhibitionism, and Entitlement/Exploitativeness (NPI-13; [40, 53]). The entitlement/exploitativeness subfactor is related to maladaptive interpersonal outcomes, whereas the leadership/authority and grandiose exhibitionism factors have been linked to more adaptive traits and behaviors. The NPI can be reliably measured with Likert-type scales in lieu of the original binary questions [54]. Each narcissistic item was rated on a 5-point Likert-type scale (1: "strongly disagree," 5: "strongly agree"). Reliability of the NPI-13 was good ($\alpha$ = .85).

**Vulnerable narcissism.** The Pathological Narcissism Inventory was developed to assess pathological narcissism and is composed of items that assess both grandiose and vulnerable characteristics [41]. The original PNI includes 7 subscales that load onto the higher order factors of grandiose narcissism (exploitativeness, self-sacrificing self enhancement, grandiose fantasies, and entitlement rage) and vulnerable narcissism (contingent self-esteem, hiding the self, and devaluing). For this study, we utilized the vulnerable subscale from a brief version of the PNI containing 6 items [42]. Each item was rated on a 6-point Likert-type scale (0: "not at all like me," 5: "very much like me"). The reliability of SB-PNI-V was acceptable ($\alpha$ = .74).

## Story ratings

All stories were rated for plot potential ("How good is the story plot") and the teller's performance ("How well the story is told") as conveyed by the teller along with the affiliative behavior exhibited by the recipient and rated by three independent observers (research assistants), unaware of the study goals. Potential was rated from the audio and performance/affiliation from the video at independent passes, with the order randomized individually for each rater. The raters listened to each story and evaluated each story plot on a scale of 1–5 (1: very little story potential; 5: very good story potential). In a separate pass, they viewed all the videos in random order and evaluated "how well the story was told" on a scale of 1–5 (1: very poor, even for an ordinary conversation; 5: excellent, for a non-professional) and the recipient's displayed affiliation (Finnish: *myötäeläminen*) on a scale of 1–5 (1: non-affiliative; 5: very affiliative). Inter-rater reliability for all variables was assessed by computing intraclass correlation coefficients based on a mean-rating (k = 3), absolute-agreement, and 2-way random-effects model [55]. In the statistical analysis, the mean values of the three raters were used.

## Analysis

First, as a manipulation check, we used a linear regression to test whether the story recipients were rated as less affiliative in the distracted recipient condition than in the non-distracted recipient condition. Here, we expected to see an above medium effect size (d > 0.5). Further preliminary analysis was conducted to confirm similar levels in plot potentials between the two recipient conditions, also by using a linear regression. The effect of story length (in minutes), dyad-level sex (male vs. female) and ages of both the teller and listener were added as covariates to control for their effects.

H1 was assessed with a linear regression model where the predicted variable was the mean ratings of the teller's performance. Between-subjects predictor was the recipient distraction (distracted vs. non-distracted), with story length, dyad-level sex and ages of both the teller and listener as covariates. We hypothesized a significant effect indicating lower teller's performance in the distracted recipient condition. Based on the original study [1], we expect a large effect size (d > 0.8) for the main effect predicting teller's performance along with non-significant sex and story length effects.

H2 was assessed with separate models where the predicted variables were teller's performance, teller's self-reported valence and arousal ratings, and physiological indices of teller's phasic SC and HR during the narration. In each model, the condition (distracted vs. non-distracted) was set as a between-subjects factor and the dyad-level sex and ages of both participants were set as covariates. For H2, the main effect of the condition was assessed. To test H3, a main effect of teller's narcissism as well as the interaction between narcissism and the distraction condition were added to the models as predictors. The models were estimated for the same outcomes as in the case of H2 and separate models were conducted for different narcissism traits (NPI-13 vs PNI-V) as moderators. For H3, we expected the interaction between

distraction and narcissism to be significant, with the effect size of the vulnerable narcissism being larger compared to grandiose narcissism. The direction of the interaction effect was determined by plotting the simple slopes.

## Results

### Preliminary analyses

Mean story length was 135.1 seconds (SD = 73.48), with no difference due to distraction, $t(40)$ = 0.51, $p$ = .615. Observer ratings exhibited moderate (story potential, ICC(2,k) = .64; teller's performance ICC(2,k) = .60) to good agreement (affiliative behavior, ICC(2,k) = .87) and were deemed satisfactory for the use of the mean scores between raters as variables in further analysis. There was one story with a mean plot potential of 1 and three stories with mean plot potential of 1.33. However, analyses were performed with the full group of participants.

To examine whether the distraction condition caused the recipients to show observable changes in the displays of affiliation towards the storytellers, a linear regression model was calculated. F-test results of the dependent variables revealed a non-significant main effect of distraction, $F(1, 36)$ = 0.013, $p$ = .908. F-test results regarding the control variables (age of the recipient, age of the teller, dyad gender, and story length) were likewise non-significant ($p$s > .177). The regression weights and standardized beta weights of the model are reported in S1 Table in S1 File. Examining the story's plot potential revealed no significant effect of distraction in F-test results, $F(1, 36)$ = 2.245, p = .143. As can be seen from S1 Table in S1 File, however, story length was positively associated with plot potential ratings, $b$ = .279, $p$ = .022. The other control variables had non-significant associations with plot potential ($p$s > .107).

### H1: Effect of recipient distraction on teller's performance

The regression model results regarding H1 are shown in Table 2. Contrary to the hypothesis according to which the recipient's distraction would reduce the performance of the tellers, no significant main effect of distraction was found, $F(1, 36)$ = 0.237, $p$ = .630. None of the control

**Table 2. Regression models predicting teller's performance with distraction (H1), teller's and recipient's age, gender, story length, and recipient's affiliation (N = 42).**

| Predictors | Teller's performance | | | Teller's performance + Affiliation | | |
|---|---|---|---|---|---|---|
| | b | beta | b 95% CI [LL, UL] | b | beta | b 95% CI [LL, UL] |
| Intercept | 3.17 *** | 0 | 1.72–4.62 | 2.03 ** | 0 | 0.54–3.52 |
| Age (Teller) | 0.02 | 0.23 | -0.01–0.04 | 0.02 * | 0.3 | 0.00–0.05 |
| Age (Recipient) | -0.01 | -0.15 | -0.04–0.01 | -0.01 | -0.12 | -0.03–0.01 |
| Gender (Dyad) | -0.14 | -0.08 | -0.68–0.41 | -0.18 | -0.11 | -0.67–0.31 |
| Story length (min) | 0.16 | 0.25 | -0.06–0.37 | 0.09 | 0.14 | -0.11–0.28 |
| Distraction | -0.11 | -0.08 | -0.59–0.36 | -0.1 | -0.07 | -0.53–0.32 |
| Recipient affiliation | | | | 0.34 ** | 0.45 | 0.12–0.56 |
| $R^2$ | | | 0.173 | | | 0.355 |

*Note*. A significant b-weight indicates that the beta-weight is also significant. b represents unstandardized regression weights whereas beta indicates standardized regression weights. LL and UL represent the lower and upper limits of 95% confidence intervals of unstandardized regression weights.

* p<0.05

** p<0.01

*** p<0.001.

variables (age, partner age, gender, story length) had significant main effects ($ps > .145$). A follow-up model was then conducted to examine whether recipient affiliation was associated with teller's performance (see Table 2, model on the right). To this end, we added the affiliation score as a covariate to the linear regression model predicting teller's performance ratings. A significant positive association was found between recipient affiliation and teller's performance, $b = .338$, $p = .003$, indicating that tellers receiving more affiliation from their recipients performed better in storytelling.

## H2: Effect of recipient's distraction on teller's affective valence and arousal

Next, H2 was tested according to which distracting story recipients would result in more negative self-reported valence ratings and increased self-reported and autonomic arousal in the tellers. In contrast to H2, distraction had no significant main effect on self-reported valence, $F(1, 36) = 0.849$, $p = .363$ (see S2 Table in S1 File for regression weights). However, a significant main effect of distraction was discovered for self-reported arousal, $F(1, 36) = 5.078$, $p = .030$), indicating that tellers, whose recipients were distracted reported significantly higher arousal ($M = 3.82$, $SD = 1.84$) compared to the control group ($M = 2.72$, $SD = 1.52$). These results, which partially supported H2, were preserved while controlling for ages of the teller and the recipient, their gender, and the story length.

Regression models regarding teller's phasic SC and HR during storytelling, are reported in S3 Table in S1 File. Contrary to H2, no significant main effect of distraction was found on phasic SC activity, $F(1, 32) = 0.600$, $p = .444$), or HR, $F(1, 33) = 0.201$, $p = .657$. The effects of controlled covariates on phasic SC were non-significant ($ps > .490$) but a main effect of recipient's age on HR was found, $F(1, 33) = 8.373$, $p = .007$, indicating a negative association between age and HR, $b = -0.605$, $p = .007$.

## H3: Moderating influence of tellers' narcissistic traits on their self-reported and autonomic affective reactions to recipients' distraction

**Performance and self-reported affect.** By testing H3 regarding moderating influences of teller's narcissism traits on the teller's performance, analyses revealed no significant interaction effects neither between NPI and distraction, $F(1, 34) = 1.137$, $p = .294$, nor between PNI-V and distraction, $F(1, 34) = 0.811$, $p = .374$, (see S4, S5 Tables in S1 File for regression coefficients). The NPI×distraction interaction effects on self-reported valence, $F(1, 34) = 0.382$, $p = .541$, and arousal, $F(1, 34) = 0.140$, $p = .711$, were likewise non-significant (see S6 Table in S1 File for regression weights). Similarly, no significant interactions between PNI-V and distraction were found on self-reported valence, $F(1, 34) = 0.163$, $p = .689$, or arousal, $F(1, 34) = 0.163$, $p = .689$ (see S7 Table in S1 File for regression weights). Therefore, no support for H3 was found when predicting the teller's performance or self-reported affect. The effects of ages of the teller and the recipient, their gender, and the story length were controlled in all the models.

### Physiology

Testing H3 regarding skin conductance (SC) revealed a significant NPI×distraction interaction, $F(1, 30) = 8.6691$, $p = .006$. In this model, also the main effect of NPI significantly influenced SC, $F(1, 30) = 6.403$, $p = .017$. The regression weights from the model are presented in Table 3 (left side). As can be seen from the regression weights and the simple slopes plotted in Fig 2, tellers' NPI had a positive association with phasic SC in the distraction condition, but in the non-distraction condition the association was negative. Because a single observation with a low NPI score and very high SC level could have driven the negative trend in the non-distraction condition, we scrutinized the finding by rerunning the model on data lacking this

**Table 3. Regression models predicting phasic CS and HR with teller's NPI score as moderator.**

| Predictors | Phasic CS | | | HR | | |
|---|---|---|---|---|---|---|
| | b | beta | b 95% CI [LL, UU] | b | beta | b 95% CI [LL, UU] |
| Intercept | 3.18 | -0.11 | -6.62–12.98 | 42.18 *** | | 18.60–65.75 |
| Age (Teller) | 0.08 | 0.17 | -0.09–0.25 | -0.06 | -0.05 | -0.45–0.34 |
| Age (Recipient) | 0.02 | 0.04 | -0.17–0.21 | -0.62 ** | -0.49 | -1.06 –-0.17 |
| Gender (Dyad) | -1.64 | -0.17 | -5.09–1.82 | -4.67 | -0.21 | -12.93–3.59 |
| Story lenght (min) | 0.43 | 0.11 | -1.17–2.03 | 0.66 | 0.07 | -3.08–4.41 |
| Distraction | -0.83 | -0.07 | -3.88–2.23 | -0.5 | -0.02 | -7.60–6.61 |
| NPI | -3.91 * | -0.29 | -7.06 –-0.75 | -2.24 | -0.20 | -9.43–4.94 |
| Distraction*NPI | 4.98 ** | 0.56 | 1.53–8.44 | 0.43 | 0.02 | -7.53–8.38 |
| N | 38 | | | 39 | | |
| R² | 0.271 | | | 0.243 | | |

*Note.* Phasic SC refers to storytelling-related phasic skin conductance activity divided by phasic skin conductance activity measured during baseline. HR refers to change in heart rate (beats per minute) as compared to the baseline HR level (HR-task—HR baseline). A significant b-weight indicates that the beta-weight is also significant. *b* represents unstandardized regression weights whereas beta indicates standardized regression weights. LL and UL represent the lower and upper limits of 95% confidence intervals of unstandardized regression weights. The NPI stands for standardized total score on 13-item narcissism personality inventory (NPI-13).

* $p < 0.05$

** $p < 0.01$

*** $p < 0.001$.

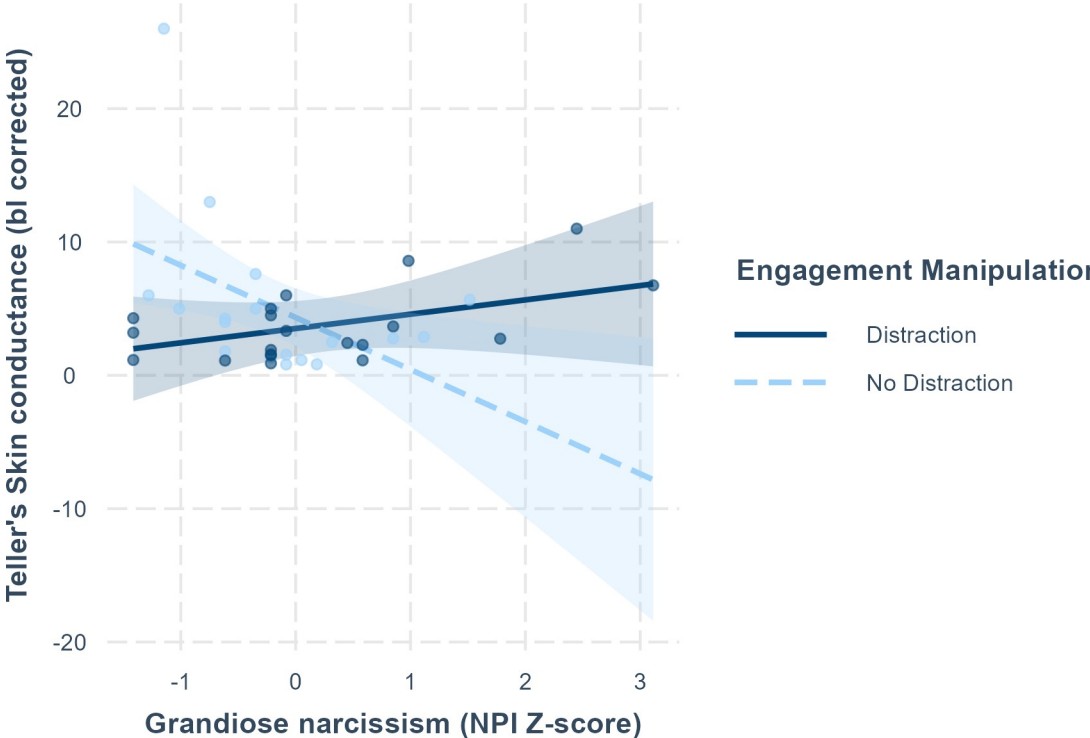

**Fig 2. Interaction between the NPI total score (standardized) recipient distraction on phasic SC relative to baseline.** The lines represent linear regression slopes (simple slopes) between NPI and Phasic SC on each level of distraction.

observation. As a result, the interaction remained significant, $F(1, 29) = 5.474$, $p = .026$, and the pattern of results preserved lending support for H3. Then, when predicting teller's HR (see Table 3, right side), we found non-significant F-test result for NPI, $F(1, 31) = 0.405$, $p = .529$, and for the NPI×distraction interaction, $F(1, 31) = 0.012$, $p = .914$, indicating that only the effect on phasic SC was moderated by grandiose narcissism.

Finally, we examined whether teller's PNI-V score would have a moderating influence on phasic SC and HR (see S8 Table in S1 File for regression weights). There were no significant main or interaction effects due to PNI-V on phasic SC ($ps > .895$). When predicting HR, however, a significant effect of PNI-V was found, $F(1, 31) = 4.328$, $p = .046$. Tellers higher in PNI-V were found to exhibit higher HR during narration as indicated by a positive beta weight of PNI-V ($b = 5.371$). The interaction between distraction and vulnerable narcissism (PNI-V×distraction) was, however, non-significant ($p = .453$), speaking against the assumption that vulnerable narcissism would have a moderating influence and/or that its moderating influence would be larger compared to that of grandiose narcissism.

## Discussion

In this study, we examined whether grandiose and vulnerable narcissism influence storyteller's behavioral, emotional, and psychophysiological reactions to story-recipient's disengagement. Here we will discuss the results in relation to prior literature and our pre-registered hypotheses. First, we hypothesized that the counting task would decrease the recipients' ability to affiliate with the storyteller. Interestingly, we found that the main effect of the distraction condition on recipients' affiliation rated by independent observers was non-significant. It is possible that our storytelling setting involving instructions displayed on computer screens and psychophysiological electrodes and cables discouraged participants to express affiliation by multimodal bodily cues. Therefore, the ratings of affiliation might have reflected the recipients' overall friendliness, for example, instead of affiliation expressed in response to specific story events. However, this finding did not prevent us from investigating our main hypotheses relating to the effects of the teller's narcissistic traits on their performance, emotional valence, and psychophysiology.

Our first hypothesis (H1) was informed by the study by Bavelas and colleagues [1], who showed that distracting a story recipient from the story content is associated with problems in the teller's performance. Our results lend no support for the hypothesis since disengagement of the recipient was not associated with a drop in the tellers' performance. However, in further exploratory analyses we did find a direct link between recipient's observed affiliation behavior and teller's performance indicating that tellers receiving more affiliation from their recipients performed generally better in the storytelling. This finding underlines the bidirectional relationship between recipient's and teller's interactional participation, which may emerge also in the presence of mental distractors.

Next, we hypothesized that distraction of the recipient is associated with lower self-reported valence and increased self-reported and autonomic arousal in the teller as compared to the non-distracted recipient condition (H2). With regard to self-reported affect, our data lend partial support for this hypothesis. The tellers, whose recipients were distracted, reported significantly higher arousal, as compared to those who told their story to non-distracted recipients. This result suggests that distraction of the recipient did affect the tellers' experience of the telling situation, causing them to be more emotionally aroused, at least subjectively. With regard to autonomic arousal, however, there was no significant effect of the distraction on either the tellers' SC activity or HR. These results did not change when recipient affiliation was included in the model as a covariate (result available upon request). This disparity between self-report

and autonomic arousal concerning telling a story to a distracted conversational partner conforms and adds to existing evidence of misalignment between self-report and physiological response [56] and calls for further research.

Finally, our third hypothesis was based on earlier evidence that both grandiose and vulnerable narcissism are associated with emotional reactivity to social rejection and self-threatening feedback. We thus hypothesized that the teller's narcissism traits would moderate the effect of distraction on the teller's performance, their self-reported affect, and their psychophysiological reactivity (H3). Our results on the tellers' performance and self-reported valence/arousal failed to support the hypothesis as neither grandiose (NPI) nor vulnerable (PNI-V) narcissism moderated the effect of recipient distraction. However, with regard to the tellers' autonomic arousal, H3 received support. The tellers higher in grandiose narcissism had higher SC when telling the close call to a disengaged recipient than when telling a story to an engaged recipient. Interestingly, this difference in SC was not present in individuals scoring low in grandiose narcissism which indicated that grandiose narcissists are especially reactive to others' disengagement in their sympathetic arousal. The same effect, however, did not occur in their HR. The lack of correspondence between obtained SC and HR results can also be partially explained due to HR being more susceptible to influence and artifacts from speech [57] and the task's possible evoking of the cardiac defense mechanism, comprised of accelerative and decelerative components [58]. Contrary to SC, which reflects pure sympathetic arousal, these components can sum up to equivalence when calculating a mean value from a relatively short duration.

Given that vulnerable narcissism has been associated with high physiological reactivity to social stressors, we assumed that tellers with high levels of vulnerable narcissism would exhibit stronger autonomic reactivity in response to recipient's disengagement than those with high levels of grandiose narcissism. Our results did not support this conclusion. The tellers higher in vulnerability, however, were found to exhibit generally higher HR during narration in both distraction and non-distraction conditions. This might be related to the result of our post-hoc observation that vulnerable narcissists were rated as worse storytellers. The vulnerable participants' reactivity to the overall stress-inducing social situation could have impacted their sensitivity to disengagement from the co-participant. The greater susceptibility to perceive the task as a stressor, as evidenced by their heightened autonomic sensitivity [17, 37] could constitute a larger influence than the effect of experimental manipulation, thus obfuscating the results. Covariates in all the models were not significant in general, except for significant effect of recipient's age on HR indicating a negative association between age and HR. This finding corresponds to the well-established pattern of task-related HR decreasing as a function of age [59].

Overall, our results support the general thesis of Bavelas and colleagues [1], as well as conversation analysts' findings over several decades (e.g. [4, 5, 60–62]) that story-recipients' actions affect the teller's conduct, as in our data tellers receiving more affiliation from their recipients performed better in storytelling. Even though in our data the independent raters' evaluation of recipient affiliation was comparable in the two conditions of distraction, the tellers' who told their stories to distracted recipients still picked up on their disengagement and reported feeling higher arousal than those whose recipients were not distracted. It is possible that the storytellers sensed their recipients' interactional disengagement in the real-time telling situation with higher sensitivity than our independent raters, who were only looking at videos after the fact. In addition, the videos that our raters evaluated recorded the tellers and recipient's faces separately, so that the bidirectional engagement between the teller and recipient was more difficult to assess, possibly leading to an evaluation of affiliation that was more focused on the individual receiving the story and not in the interactional events occurring between the two parties.

Our most important finding concerns the connection between the tellers' grandiose narcissistic traits and their elevated SC responses to a distracted story-recipient. This arousal effect did not show in the self-report measures, which is in line with studies that have shown that narcissists do not always report emotional distress concurrent with physiological reactions [11, 22, 23]. This finding is also in line with our hypothesis that narcissistic individuals are more sensitive to face threats in interaction, which can show in their bodies as physiological arousal. We assumed, however, that this effect would be bigger with vulnerable narcissism than grandiose narcissism, which turned out not to be the case. Vulnerability did not have a moderating influence on the tellers' autonomic arousal. This calls to question whether the distinction between the two narcissism types was ideal for our purpose. More recent studies have suggested a trifurcated model of narcissism [63–65], where the dichotomy of grandiose and vulnerable narcissism is replaced with combinations of the Five-Factor-Model trait variants which form three facets: agentic, antagonistic, and neurotic narcissism [66]. If we would have examined the tripartite model, we would have expected to see the reactivity in the antagonistic factor, since it is common to both grandiosity and vulnerability. Antagonism may also better untangle vulnerability because vulnerability is largely related to neuroticism. We originally decided to use the NPI and PNI questionnaires in our study, since their use is well established, they have been translated in Finnish, as well as validated. Future studies are needed to test whether the trifurcated model produces similar results.

Our study has at least four key limitations. First, we were not able to reach the planned sample size stated in the preregistration and the obtained sample of storytellers used in the hypothesis testing was modest. Consequently, the study could provide sufficient statistical power to detect only relatively large effects. It is possible that some of the hypothesized effects were not detected due to the underpowered sample. Then, regarding type-I error, the division of narcissistic traits to two dimensions and the large total number of outcome measures resulted in a relatively large number of statistical models, which could have increased the risk of the type-I error. Further studies are, therefore, required to examine replicability of the obtained findings. Second, compared to rating of affiliative behavior and the indices reported in the original study by Bavelas and colleagues [1], there was more disagreement between raters in rating of story potential and teller's performance, as indicated by smaller, though still sufficient, internal reliability indices. This suggests that more effort should be concentrated on rater instruction in the future studies. Third, due to experimental constraints, brief versions of the measures were used to operationalize narcissism. Usage of full-length versions and/or additional scale instruments could potentially increase the psychometric accuracy in measuring the underlying trait(s) of interest. Fourth, the sample was composed of young and mostly female university students which affects the generalizability of the results to the total population. Furthermore, the sample may not have been ideal for examining narcissistic pathology, as there is evidence that while grandiose and vulnerable narcissism are unrelated at low-to-moderate levels of grandiose narcissism, they are related at high levels of grandiose narcissism [66]. Thus, there can be special physiological patterns regarding individuals with clinically relevant narcissism. Future studies using clinical samples and more rigorous narcissism and psychopathological constructs are therefore required. These limitations notwithstanding, there were multiple strengths in the current study. Indeed, earlier research on narcissism has relied heavily on self-reports, whereas we included objective raters' evaluations and psychophysiological measures in our examination. All the tested models were preregistered, which speaks for the robustness of the obtained effects. Moreover, considering that the naturalistic dyadic experimental design was likely to increase the proportion of unexplained variance in the outcome measures, the obtained interaction effect between narcissism and disengagement on autonomic arousal is notable.

To conclude: the grandiose narcissists' increased arousal showed in their psychophysiology but not in their self-reports. This is consistent with the literature on narcissism and 'repressors' [11, 31, 67], individuals who utilize a repressive coping style tend to avoid negative affect and entertain illusions of unrealistic optimism and overly positive self-evaluations. Our results also enforce an interactionist view of personality [68], where manifestations of personality can be seen as a function of the person and the social situation. The task of this view has been to find out the consistency of people's behavior in different situations. Our study shows that when narcissistic individuals 'self' is at stake, we can make more

accurate predictions about their behavior. Finally, our study is in line with recent accounts that grandiose narcissists' ability to read other people's emotions is intact, and the question is more about their lack of motivation to empathize [69, 70], as our experiment showed that grandiose narcissists are emotionally sensitive to their co-participants disengagement.

## Supporting information

**S1 File.**
(DOCX)

## Acknowledgments

The authors wish to thank Sanna Kie Kettunen for her invaluable role in the data collection, as well as other members of the 'Facing Narcissism' group for their insightful input during different stages of the project.

## Author Contributions

**Conceptualization:** Emmi Koskinen, Pentti Henttonen, Ville Harjunen, Elizabeth Krusemark, Anssi Peräkylä.

**Formal analysis:** Pentti Henttonen, Ville Harjunen.

**Funding acquisition:** Anssi Peräkylä.

**Investigation:** Emmi Koskinen, Mariel Wuolio.

**Methodology:** Pentti Henttonen, Ville Harjunen, Elizabeth Krusemark.

**Project administration:** Emmi Koskinen, Mariel Wuolio.

**Supervision:** Anssi Peräkylä.

**Writing – original draft:** Emmi Koskinen, Pentti Henttonen, Ville Harjunen, Elizabeth Krusemark, Matias Piispanen, Liisa Voutilainen, Mariel Wuolio, Anssi Peräkylä.

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
