## [Decision Letter · Decision Letter 0]

6 Mar 2024

PONE-D-24-01565Putting self at stake by telling a story: Storyteller’s narcissistic traits modulate physiological emotional reactions to recipient’s disengagement

PLOS ONE

Dear Dr. Koskinen,

Thank you for submitting your manuscript to PLOS ONE. After careful consideration, we feel that it has merit but does not fully meet PLOS ONE’s publication criteria as it currently stands. Therefore, we invite you to submit a revised version of the manuscript that addresses the points raised during the review process.

We look forward to receiving your revised manuscript.

Kind regards,

Tzen-Yuh Chiang

Academic Editor

PLOS ONE

“This work was supported by the Research Council of Finland, project numbers 319113 and 320248 (https://www.aka.fi/en/), and the Society of Swedish Literature in Finland (https://www.sls.fi/en).”

Reviewers' comments:

Reviewer's Responses to Questions

**Comments to the Author**

1. Is the manuscript technically sound, and do the data support the conclusions?

Reviewer #1: Yes

2. Has the statistical analysis been performed appropriately and rigorously? 

Reviewer #1: Yes

3. Have the authors made all data underlying the findings in their manuscript fully available?

Reviewer #1: Yes

4. Is the manuscript presented in an intelligible fashion and written in standard English?

Reviewer #1: Yes

5. Review Comments to the Author

Reviewer #1: This manuscript explores a conversational experiment where participants tell a personally meaningful story ("close call") to an unknown co-participant. The co-participant is instructed to either listen to the story intently, or is given a mental computational (distraction) task. Therefore in the two conditions the level of interactional engagement is systematically modified. Explorations between these two conditions were conducted regarding the quality of interaction between the participants, subjective ratings of emotional arousal, and psychophysiological activity. Importantly, the authors were also specifically interested in any associations between narcissistic features (grandiosity/vulnerability) and the key outcome variables under examination, and indeed they reported some associations between narcissistic functioning and elevated interpersonal hypersensitivity - which is noteworthy.

Overall, due to the experimental design and focus on narcissistic pathology the study is highly novel, it asks an important scientific question with hypotheses that follow clearly from prior research, it is well written, its methods were rigorous and well described and the findings (including non-significant findings) were meaningfully discussed. Despite a number of the hypotheses not being supported, as well as some inherent limitations regarding design of the study, this manuscript provides a meaningful contribution to the literature that likely can serve as a template for future research and experimental designs to extend and improve on. This is the basis of my recommendation for the article.

Notwithstanding, there are a few points the authors may wish to consider in a subsequent revision:

- Being that a key variable in this study is narcissistic pathology, it was somewhat surprising the very brief narcissism measures were used to operationalize this element. As a result, it is reasonable to quite seriously question the robustness of the narcissism variable under examination and any conclusions to be drawn (particularly if there were more significant findings). Further, the two expressions of narcissism were measured using scales that are somewhat contradictory in their theoretical assumptions (i.e., 'healthy/adaptive' and 'pathological' expressions). While there is some discussion of this regarding the use of the Five-Factor-Model trait variants, but I think more could be made of this limitation, which is quite central to the design of the study.

- A similar limitation relates to the sample under examination being university students. While quite a common convenience sample to select, it is not ideal for examining narcissistic pathology as scores in published literature using such samples tend to be quite low - which again raises question to what extent are we truly examining narcissistic pathology. Again, there is a comment on this in the limitations, however this could be expanded. While of course nothing can be done in the current design to account for this, the authors could comment on possible experimental designs that would allow for a more robust examination, using clinical samples and more rigorous narcissism and psychopathological constructs to help guide future research which may like to build on this study.

- Finally, while some participant demographics and measures descriptives are variously included in text throughout, the manuscript would benefit from a table summarizing these features in order to assist readers to understand the sample under examination better.

Thankyou for the opportunity to review this interesting manuscript.

6. PLOS authors have the option to publish the peer review history of their article (what does this mean?). If published, this will include your full peer review and any attached files.

Reviewer #1: **Yes: **Nicholas Day

---

## [Author Response · Author response to Decision Letter 0]

26 Mar 2024

PLOS ONE: Response to Reviewers

Dear Editor and Reviewers of PLOS ONE,

We thank you for the valuable feedback and useful suggestions to improve our manuscript. In this memo we respond to the comments and questions raised to the best of our abilities.

Reviewer #1:

Notwithstanding, there are a few points the authors may wish to consider in a subsequent revision:

- Being that a key variable in this study is narcissistic pathology, it was somewhat surprising the very brief narcissism measures were used to operationalize this element. As a result, it is reasonable to quite seriously question the robustness of the narcissism variable under examination and any conclusions to be drawn (particularly if there were more significant findings). Further, the two expressions of narcissism were measured using scales that are somewhat contradictory in their theoretical assumptions (i.e., 'healthy/adaptive' and 'pathological' expressions). While there is some discussion of this regarding the use of the Five-Factor-Model trait variants, but I think more could be made of this limitation, which is quite central to the design of the study.

Thank you for this insightful comment. Narcissism research has identified unique, but overlapping dimensions of narcissism (e.g., grandiose and vulnerable narcissism) that incorporate traits and behaviors that can be viewed as respectively adaptive (e.g., grandiosity, high self-esteem, extraversion) and pathological (e.g., vulnerability, low self-esteem, increased psychopathological symptoms). Despite their distinctions, both aspects of grandiose and vulnerable narcissism are related to psychopathological and interpersonal outcomes (e.g., both are related to Narcissistic Personality Disorder (NPD), relationship difficulties, and elevated reactivity to socio-evaluative stressors). Recent research has also noted that these manifestations of narcissism may best be captured using multiple narcissism measures or more recently developed (and less established) multi-factor narcissism measures. Our decision to use the NPI and the PNI was based on a review of previous research demonstrating their respective capabilities to capture the grandiose and vulnerable dimensions. Moreover, brief versions of these measures were recently validated in a Finnish sample (see Henttonen et al., 2022, and comments below). Finally, we wanted to include multiple narcissism dimensions while keeping the total experiment duration to a minimum and keep the ratio of time used for the paradigm and form-filling reasonable. However, we carefully examined and decided on the selected measures prior to the experiment.

To our understanding, the brief versions of NPI (Gentile et al., 2013) and PNI (Schoenleber et al., 2015) have demonstrated convergent and divergent validity comparable to the original versions using adequately sized samples among clinical and community participants. In a separate prior validation study (n=439, Henttonen et al., 2022), using a comparable sample, the translated brief versions of the questionnaires were observed to demonstrate good psychometric properties in general and in relation to other brief measures of narcissism (HSNS, g-FLUX). Affirming previous findings, NPI-13 accounted for the greatest amount of variance in self-reported PDQ-NPD symptoms and conformed to the expected pattern of higher self-esteem and well-being, while SB-PNI Vulnerability exhibited the largest predictive power for diminished self-esteem, well-being and psychopathology.

The references to the original and brief versions of the scales and the Finnish validation study are included in the Method section (p. 10), whereas rationale concerning the contemporary debate on grandiose-vulnerable vs three-facet dimensions of narcissism is included in the discussion section in the original manuscript. However, we have added an additional statement to the limitations section (page 18): “due to experimental constraints, brief versions of the measures were used to operationalize narcissism. Usage of full-length versions and/or additional scale instruments could potentially increase the psychometric accuracy in measuring the underlying trait(s) of interest.”

- A similar limitation relates to the sample under examination being university students. While quite a common convenience sample to select, it is not ideal for examining narcissistic pathology as scores in published literature using such samples tend to be quite low - which again raises question to what extent are we truly examining narcissistic pathology. Again, there is a comment on this in the limitations, however this could be expanded. While of course nothing can be done in the current design to account for this, the authors could comment on possible experimental designs that would allow for a more robust examination, using clinical samples and more rigorous narcissism and psychopathological constructs to help guide future research which may like to build on this study.

In this study we adopted a continuum view of narcissism where individuals are seen to exhibit traits of grandiose and vulnerable narcissism in varying degrees, ranging from low levels to pathological levels. According to this view, studying participants with relatively low levels of these traits can still provide insight on the phenomenon of narcissism in general. We have also tried to select our scale instruments carefully in order to also include pathological aspects of narcissism (see the comment above). However, as there is also evidence that while grandiose and vulnerable narcissism are unrelated at low-to-moderate levels of grandiose narcissism, they are related at high levels of grandiose narcissism (Jauk et al. 2022) and thus there can be special physiological patterns regarding individuals with clinically relevant narcissism. We have now expanded the limitations section to include this information, as well as made suggestions for future studies according to the reviewer’s suggestion (see page 19).

- Finally, while some participant demographics and measures descriptives are variously included in text throughout, the manuscript would benefit from a table summarizing these features in order to assist readers to understand the sample under examination better.

We have included full information of the participant demographics, per condition and role, regarding age and NPI/PN sum scores in the new Table 1 (page 7). The table also depicts the experimental measures, per condition, of rated story quality, teller performance, recipient affiliation, PANAS valence, arousal and dominance. Other table identifiers have been modified accordingly.

---

## [Decision Letter · Decision Letter 1]

10 Apr 2024

Putting self at stake by telling a story: Storyteller’s narcissistic traits modulate physiological emotional reactions to recipient’s disengagement

PONE-D-24-01565R1

Dear Dr. Koskinen,

We’re pleased to inform you that your manuscript has been judged scientifically suitable for publication and will be formally accepted for publication once it meets all outstanding technical requirements.

Kind regards,

Tzen-Yuh Chiang

Academic Editor

PLOS ONE

Additional Editor Comments (optional):

Reviewers' comments:

Reviewer's Responses to Questions

**Comments to the Author**

1. If the authors have adequately addressed your comments raised in a previous round of review and you feel that this manuscript is now acceptable for publication, you may indicate that here to bypass the “Comments to the Author” section, enter your conflict of interest statement in the “Confidential to Editor” section, and submit your "Accept" recommendation.

Reviewer #1: All comments have been addressed

2. Is the manuscript technically sound, and do the data support the conclusions?

Reviewer #1: (No Response)

3. Has the statistical analysis been performed appropriately and rigorously? 

Reviewer #1: (No Response)

4. Have the authors made all data underlying the findings in their manuscript fully available?

Reviewer #1: (No Response)

5. Is the manuscript presented in an intelligible fashion and written in standard English?

Reviewer #1: Yes

6. Review Comments to the Author

Reviewer #1: The changes made to the manuscript improve its clarity and utility as a piece of scientific literature, namely the inclusion of key limitations, suggestions for future research and descriptive data on key measures (so readers will have a clearer picture of the sample under examination, which is important). I thank the authors for their response and for providing necessary additions and modifications.

7. PLOS authors have the option to publish the peer review history of their article (what does this mean?). If published, this will include your full peer review and any attached files.

Reviewer #1: **Yes: **Nicholas Day

---

## [Editor Report · Acceptance letter]

11 May 2024

PONE-D-24-01565R1 

PLOS ONE

Dear Dr. Koskinen, 

I'm pleased to inform you that your manuscript has been deemed suitable for publication in PLOS ONE. Congratulations! Your manuscript is now being handed over to our production team.

Kind regards, 

on behalf of

Dr. Tzen-Yuh Chiang 

Academic Editor

PLOS ONE